# Applicability of Actigraphy for Assessing Sleep Behaviour in Children with Palliative Care Needs Benchmarked against the Gold Standard Polysomnography

**DOI:** 10.3390/jcm11237107

**Published:** 2022-11-30

**Authors:** Larissa Alice Kubek, Patrizia Kutz, Claudia Roll, Boris Zernikow, Julia Wager

**Affiliations:** 1PedScience Research Institute, 45711 Datteln, Germany; 2Department of Children’s Pain Therapy and Paediatric Palliative Care, Faculty of Health, School of Medicine, Witten/Herdecke University, 58455 Witten, Germany; 3Department of Neonatology, Pediatric Intensive Care and Sleep Medicine, Children’s and Adolescents’ Hospital, Witten/Herdecke University, 45711 Datteln, Germany; 4Palliative Care Centre, Children’s and Adolescents’ Hospital, 45711 Datteln, Germany

**Keywords:** chronic disease, palliative care, actigraphy, polysomnography

## Abstract

In children with life-limiting conditions and severe neurological impairment receiving pediatric palliative care (PPC), the degree to which actigraphy generates meaningful sleep data is uncertain. Benchmarked against the gold standard polysomnography (PSG), the applicability of actigraphy in this complex population was to be assessed. An actigraph was placed on N = 8 PPC patients during one-night polysomnography measurement in a pediatric tertiary care hospital’s sleep laboratory. Patient characteristics, sleep phase data, and respiratory abnormalities are presented descriptively. Bland-Altman plots evaluated actigraphy’s validity regarding sleep onset, sleep offset, wake after sleep onset (WASO), number of wake phases, total sleep time (TST) and sleep efficiency compared to PSG. PSG revealed that children spent most of their time in sleep stage 2 (46.6%) and most frequently showed central apnea (28.7%) and irregular hypopnea (14.5%). Bland-Altman plots showed that actigraphy and PSG gave similar findings for sleep onset, sleep offset, wake after sleep onset (WASO), total sleep time (TST) and sleep efficiency. Actigraphy slightly overestimated TST and sleep efficiency while underestimating all other parameters. Generally, the Actiwatch 2 low and medium sensitivity levels showed the best approximation to the PSG values. Actigraphy seems to be a promising method for detecting sleep problems in severely ill children.

## 1. Introduction

Children with life-limiting, primarily neurological conditions, such as those cared for in pediatric palliative care, commonly experience a variety of symptoms, which include sleep problems [1,2,3,4]. Unlike adult palliative care patients, the course of disease in children usually extends over many years, is characterized by numerous fluctuations between stable and unstable phases of the condition, and affects the whole sensitive family system [5,6,7,8]. 

Against this background, successful diagnosis of pertinent symptoms represents a mainstay of palliative care to alleviate children’s potential suffering and enhance their quality of life [1,9]. Many pediatric palliative care patients experience severe neurological impairment (SNI), which can be defined as follows: “Severe Neurological Impairment describes a group of disorders of the central nervous system which arise in childhood, resulting in motor impairment, cognitive impairment, and medical complexity, where much assistance is required with activities of daily living. The impairment is permanent but can be progressive or static [10]”. Children’s non-verbality, which often accompanies SNI, complicates diagnostic measures, necessitating the use of other means instead of self-report responses [1,11]. 

Even though parents become experts regarding their children over time and can provide highly valuable information about their child’s current situation, a solely subjective diagnostic approach, as in the form of third-party judgments, may not be sufficient to obtain a comprehensive, reliable picture of symptoms [2]. For example, in the case of existing sleep problems, parental claims may be biased when asked about the duration or number of waking episodes or special events during the night since parents themselves do not necessarily spend every night with their children [12]. In addition, certain parameters such as sleep stages or sleep-related breathing disorders, which are often crucial for finding a specific diagnosis, cannot be reliably recorded with subjective diagnostics. Therefore, polysomnography (PSG) has been established as the “gold standard” of sleep medicine for many decades [13,14,15,16]. Simply put, PSG uses a wide variety of objective physiological techniques such as an electroencephalogram (EEG; measuring brain activity), an electromyogram (EMG; measuring muscle activity), and an electrocardiogram (ECG; measuring heart rate), which together provide a comprehensive picture of a child’s sleep [17]. However, it also implies disadvantages that shall not be underestimated: PSG requires a night in an unfamiliar artificial environment, which may cause discomfort and stress for the child, and therefore possibly is also not representative of the sleep that is shown in the home setting. In addition, PSG is expensive, technically difficult, and not necessarily available in all medical centers [13,18,19,20].

A promising way to objectively assess pediatric sleep cost-effectively, also in the home setting, is actigraphy using actigraphs. These are portable devices comparable to a wristwatch, which record movements over a certain period of time with the help of a highly sensitive accelerometer, and can thus also differentiate between sleep and wakefulness with the help of integrated logarithms [13,18,21,22]. In recent years, actigraphy has been increasingly used in clinical as well as scientific settings [22]. However, in order to be able to test its validity and thus the significance of the data generated, a direct comparison with the gold standard PSG is necessary. In practice, this has, for example, already been investigated in samples of healthy children and adolescents and those with different underlying diseases such as Down Syndrome, Cerebral Palsy, Epilepsy, Autism, Angelman syndrome, or Craniopharyngeoma with heterogeneous results [23,24,25,26,27,28,29,30]. To the best of our knowledge, what is missing to date is the comparison of actigraphy and PSG in children with various life-limiting conditions with SNI. The potential finding that actigraphy provides reliable data for this patient population could, in the future, help to expand the possibilities of objective sleep diagnostics for these highly complex children and complement costly and burdensome laboratory examinations such as PSG. Therefore, the aim of this pilot experimental study was to compare the designated objective measures in severely impaired children. In addition, information on the children’s sleep stages and possible respiratory problems, which can only be assessed by PSG, were also analyzed in this study since the state of knowledge in this patient population is also very scarce concerning these parameters.

## 2. Materials and Methods

### 2.1. Participants

Potential participants were recruited through the pediatric palliative care unit of a pediatric tertiary care hospital. All children with a life-limiting condition and SNI, for whom an examination in the sleep laboratory of the same hospital was planned as part of the mandatory diagnostics, were eligible for the study (presupposing parental informed consent). It was irrelevant for what exact reason a PSG investigation was scheduled. Reasons for exclusion were an acute biological or psycho-social crisis in the family (e.g., death of a close relative), children reaching the acute terminal stage of life and parental insufficient local language proficiency.

A small number of cases was considered appropriate for sampling, considering the intricate conditions of study inclusion, respectively, the small achievable population. For its definition, studies with a similar design and children with complex underlying diseases were considered, ranging from N = 9–11 children [28,31,32], so that the target number of children for this study was set at approximately N = 10.

Ethical approval was granted by the Ethics Committee of Witten/Herdecke University (approval code: 128/2020, approval date: 15 August 2020). All families provided informed consent to participate in the study.

### 2.2. Actigraphy

For the actigraphy measurement, an Actiwatch 2 (Philips Respironics, Murrysville, PA, USA) was placed around the wrist of the participating children. Since a dominant hand could not be reliably determined in the children due to their underlying diseases, the recording was performed on the left or right wrist by default. The system’s accompanying Actiware software package (Philips Respironics, Murrysville, PA, USA; version 6.1.2) was used for the device’s configuration and the subsequent data scoring. Activity data were condensed into 15-s epochs. Extracted variables were sleep onset time (defined as the first immobile minutes within a set rest interval; see data analysis), sleep offset time (defined as the last 10 immobile minutes within a set rest interval; see data analysis), duration of wake time during the night (wake after sleep onset; WASO), the total number of wake episodes during the night, total sleep time (TST), and sleep efficiency (defined as the amount of time a child actually spends asleep while in bed at night; specified in percent). 

### 2.3. Polysomnography (PSG)

Polysomnography was performed in a darkened room, with the patient sleeping in a comfortable bed. The polysomnography data were obtained using Sleepware G3 Version 3.9.5 (Philips Respironics). The Standard polysomnography setup consisted of 15-channel electroencephalography (EEG; F4/C4, F3/C3, C4/P4, C3/P3, T4/Cz, T3/Cz, C4/M1, C3/M2, F3/m2, F4/M1, T3/M2, T4/M1, Cz/M2, P3/M2, P4/M1), submental electromyography (EMG), electrooculography (EOG; right/left); electrocardiogram (ECG), audio and video recording, and a standardized protocol recorded by an experienced nurse. For assessing potential respiratory events, arterial oxygen saturation by pulse oximetry (SaO_2_), transcutaneously measured pCO_2_, and airflow monitored with a nasal cannula attached to a pressure transducer, and a thermistor, thoracic and abdominal inductive plethysmography was used.

The PSG report comprises all parameters also assessed by actigraphy. Additional parameters generated by the PSG exclusively are information on the children’s sleep stages, arousals, snoring, apnoe-hypopnoe index, and desaturation index. Sleep follows a unique architecture in which rapid eye movement (REM) sleep rhythmically alternates with non-rapid eye movement (NREM) sleep. Three sleep stages with progressively increasing sleep depth are assigned to NREM [33].

In parallel with the PSG data collection, the night nurse on duty kept a handwritten log of the night’s events (e.g., the time the child fell asleep and the start of recording PSG).

### 2.4. Data Collection

Data collection took place from January 2021 to March 2022. Once it was decided that a patient from the palliative care ward should be scheduled for investigation in the sleep laboratory of the same hospital, the study coordinator (L.A.K.) was informed. In order to minimize the burden on the families, no PSG appointments were made specifically for the study purpose, but rather use was made of appointments as part of the patients’ ongoing treatment needs. Parents were provided with comprehensive verbal and written information about the study. In this process, they were also shown the actigraph to get a better picture of the unfamiliar device. If participation was desired, the configured actigraph was brought to the sleep laboratory by the study coordinator on the day of the PSG examination to be put on the child by the nurses on duty. No specific time was defined for this; the actigraph should be put on when the PSG was also placed. Both measurements were hence performed during one night. The day after the examination, the data of the PSG were read out and evaluated. The actigraph was collected for data readout and analysis.

### 2.5. Data Analysis

For the analysis of the actigraphy data, Actiware was used. For determining the rest intervals (periods in which the patient is less active and probably resting), the handwritten PSG protocols were referred to. The start of a rest interval was thus determined by calculating back 30 min from the start of sleep indicated in the protocol (for example, the protocol indicates 22:00 as the sleep onset; thus, the onset of the actigraphy rest interval is 21:30). The end of a rest interval was calculated according to the same logic (sleep end according to PSG protocol plus 30 min). Once a rest interval was defined, sleep and wake intervals within it were automatically detected and output by the Actiware. The 30-min tolerance range rule applied has been reported in the literature and was necessitated in this study for finding a reference point for the onset of actigraphy measurement [34]. For the determination of the wake phases, the four sensitivity specifications provided by Actiware were calculated. Each of these specifications requires a different number of activity counts per epoch to be considered “awake” (low sensitivity: 80 counts, medium sensitivity: 40 counts, high sensitivity: 20 counts, automatic sensitivity: counts are defined by Actiware based on a patient’s individual activity level). No prior decision was made on a particular specification as there is no evidence in the literature or practice to date on which level to choose for this particular population.

Following the acquisition, PSG data were manually scored by a professional somnologist in accordance with the AASM Manual for the Scoring of Sleep and Associated Events [35]. The result of this analysis, respectively, the sleep parameters of interest, were then passed on to the study coordinator.

Descriptive statistics were used for patient and sleep parameter specifications. The agreement between the 2 diagnostic measures was assessed using Bland-Altman plots. The Bland-Altman diagram graphically depicts the agreement or bias between 2 diagnostic measures. The *x*-axis shows the mean value of the 2 measurement methods, and the *y*-axis shows the difference between them. In addition to the mean value of the differences between both measurement methods, which is entered as a horizontal bar in the diagram, an upper and lower limit are formed using the mean and standard deviation of the difference between two tested measures and indicate the range in which 95% of the difference of the second method (actigraphy in this case) compared to the gold standard (PSG) should fall. The upper and lower limit are also plotted as horizontal bars above and below the mean of the differences between the 2 focused measurement methods. If values are within the 95% confidence range, the second method can be said to have sufficient accuracy compared to the first [36].

As recommended in the literature, the determination of the correlation coefficient was omitted for the evaluation of our study in view of the consideration that a high correlation does not automatically indicate a good agreement between two diagnostic measures and thus may be inadequate [36]. 

A prerequisite for the Bland-Altman plot is a normal distribution of the difference between the two measures and was therefore tested for our study using the Kolmogorov–Smirnov test. For the Bland-Altman plots of parameters sleep onset time and sleep offset time, the corresponding time data were converted to numerical numbers and plotted accordingly. All analyses were conducted via SPSS (version 28).

## 3. Results

A total of N = 8 children with life-limiting neurological conditions and a median age of 5.66 years (range: 1–13 years) were included in the study (*n* = 3, 37.5% female; *n* = 5, 62.5% male). Further patient characteristics are shown in Table 1. 

### 3.1. Polysomnography (PSG)

Children spent an average of 3.6% (median: 1.8, range: 0–11.2, SD: 4.4) of their sleep on average in sleep stage 1, 46.6% (median: 45.8, range: 22.2–77.9, SD: 16.2) in sleep stage 2, 38.4% (median: 39.25, range: 19.5–55.7, SD: 11.76) in sleep stage 3 and 11.3% (median: 8.1, range: 1.7–24.4, SD: 8.7) in the REM sleep phase.

Regarding respiratory parameters, children showed an average of 66.1 (median: 17.5, range: 5–338, SD: 113.6) respiratory events that lasted on average 12.9 min (range: 7–18.3, SD: 4). In detail, the most frequent conspicuities were central apneas and hypopneas (Figure 1).

### 3.2. Polysomnography (PSG) Compared to Actigraphy

Descriptively, the children’s average time of falling asleep according to actigraphy was 21:20 (median: 21:18, range: 20:26–22:02, SD: 0:31) and 21:18 (median: 21:44, range: 19:26–21:59, SD: 0:55) according to PSG. Patients awoke on average at 4:51 (median: 4:55, range: 4:01–5:40, SD: 0:33) according to actigraphy and PSG (median: 4:52, range: 4:06–5:23, SD: 0:23). 

Figure 2 shows the mean values of the various parameters for the actigraphy and PSG measurements. Regarding WASO, the mean actigraphy measurements at all sensitivity levels were descriptively lower than the mean PSG value of 79 min (median: 88.7, range: 8–154, SD: 49.2). A similar pattern emerged toward the frequency of nocturnal wake phases. For sleep efficiency, the actigraphy medium, high, and automatic sensitivity levels achieved descriptively slightly higher mean scores than the PSG (medium sensitivity, median: 84.4, range: 70.8–94., SD: 8.1; high sensitivity, median: 87.3, range: 76.4–94.6, SD: 6.6; automatic sensitivity, median: 89.7, range: 81.7–95, SD: 4.6; PSG, median: 78.9, range: 64.4–98.1, SD: 11.3). 

The differences between the two measures was calculated and tested for normal distribution. Only for the parameter sleep offset, the normal distribution assumption had to be rejected (*p* < 0.05). Bland-Altman plots are nevertheless shown for this parameter. 

Overall, the Bland-Altman plots of all studied parameters showed sufficient accuracy of actigraphy compared with the gold standard PSG. For the number of wake phases at high and automatic sensitivity, one child each fell outside the defined tolerance ranges (Figure 3).

Negative mean differences (middle horizontal bar) between the two measurement methods were evident for the parameters TST (low sensitivity: −24.62; medium sensitivity: −45.03; high sensitivity: −57.03; automatic sensitivity: −69.28), and for three settings of sleep efficiency (medium sensitivity: −2.15; high sensitivity: −4.55; automatic sensitivity: −6.99), suggesting overestimation of the parameters by actigraphy. For all other parameters, actigraphy provided lower average values than PSG and thus underestimated the gold standard’s measurement.

Regarding the different sensitivity levels of the individual parameters, it was shown that for WASO, the mean sensitivity indicated the lowest mean difference between actigraphy and PSG measurement (M = 41.62), thus representing the best setting (low sensitivity: M = 65.87; high sensitivity: M = 53.62; automatic sensitivity: M = 65.87). For the frequency of nocturnal awakenings (low sensitivity: M = 7.25; medium sensitivity: M = 7.5; high sensitivity: M = 14; automatic sensitivity: M = 24.5), TST (low sensitivity: M = −24.62; medium sensitivity: M = −45.03; high sensitivity: M = −57.03; automatic sensitivity: M = −69.28), and sleep efficiency, (low sensitivity: M = 1.58; medium sensitivity: M = −2.15; high sensitivity: M = −4.55; automatic sensitivity: M = −6.99) in each case the low sensitivity level showed the smallest mean difference between actigraphy and PSG.

## 4. Discussion

The purpose of this study was to examine the extent to which actigraphy can provide meaningful sleep data, compared with the gold standard polysomnography, in an extremely complex sample of children with life-limiting neurological conditions. In addition, descriptive PSG information on the sleep architecture and possible breathing problems of the young patients were to be contemplated.

Regarding all these aspects, to the best of the authors’ knowledge, there is no study in a comparable sample including severely ill children with severe neurological impairment to date, so for any comparisons of our results with the existing literature, it is only possible to refer to patient groups with approximately similar diseases. With regard to the polysomnography sleep stages, a comparison with children with mental retardation showed that the patients we studied basically had a similar sleep architecture with similarly distributed sleep stages in percentage. In detail, our patients spent more time in sleep stages 1–3 than those with mental retardation but less in the REM sleep stage [32]. A comparison with a sample of children with epilepsy shows a different picture. In contrast to these, our sample spent a larger proportion in stage 3 and, in contrast, fewer proportions in the remaining sleep stages [37]. Also, in comparison with children with Angelman or Prader-Willi syndrome, our findings can only be transferred to a limited extent in view of divergent distributions of sleep stages [38,39]. However, the sleep parameters in all these studies and in our study appear to differ from those of healthy children [32,37]. At this point, further studies with children with different clinical pictures are urgently needed to find out whether and which exact features the sleep architecture of severely ill children actually shows. This is of utmost importance because, until today, it is often pointed out that sleep problems are frequent in this population, but especially physiological findings are still extremely scarce [40,41,42]. More specifically, according to the International Classification of Sleep Disorders, various sleep disorders are associated with a change in sleep architecture. For example, sleep-related hypoventilation may be associated with decreased sleep stages 3 and REM, disorders of arousal with altered microstructure associated with increased slow wave activity during NREM sleep cycles [43]. However, accurate diagnosis and interpretation of these sleep disorders requires knowledge of which “baseline” changes in sleep architecture may be associated with various underlying diseases and thus can be distinguished from changes that are “purely” associated with corresponding sleep disorders [44]. Such knowledge allows not only optimization of diagnosis but also of therapy for children with life-limiting neurological conditions since a more targeted decision can be made as to which sleep disorders can potentially be cured and which can merely be regulated to an acceptable level.

Apneas and hypopneas were the most common respiratory abnormalities in our sample. This finding fits with existing literature, which also showed a predominance of these abnormalities over other respiratory problems [13,45,46,47]. Breathing problems are a common symptom, especially at the end of life in critically ill children. Diagnosis as early and conclusively as possible can help alleviate the corresponding suffering in patients and families [48]. Knowledge of which respiratory problems occur and how frequently in this population also enables the anticipatory development of different therapy strategies and may thus substantially improve the quality of life of children with life-limiting neurological conditions.

Encouragingly, regarding the polysomnography and actigraphy comparison, all studied sleep parameters except two sensitivity levels of actigraphy showed sufficient accuracy of this diagnostic measure in comparison with polysomnography. On the one hand, this indicates that actigraphy is not only an economical and easy-to-use tool but can also generate reliable sleep data in a highly vulnerable population like that of children with life-limiting neurological conditions. 

On the other hand, actigraphy has its limitations with regard to important parameters such as sleep stages or breathing problems, but actigraphy can make a valuable contribution to answering questions such as whether a child shows a disturbed sleep-wake pattern, a permanently disturbed falling asleep or waking up behavior. 

Considering that disorders of falling and staying asleep, as well as sleep-wake rhythm disorders, are among the most common disorders in children with life-limiting neurological conditions [49,50,51,52,53], this circumstance is even more valuable for the effective diagnosis and therapy of the affected children. 

Although it seems reasonable that sleep disorders in children with life-limiting neurological conditions are mainly caused by primary disruptions of sleep-regulating processes in the brain associated with the underlying disease, it should not be neglected that they may (also) be promoted by other factors such as environmental influences like noise, or inadequate sleep hygiene [11,54,55,56]. Especially in the home setting, actigraphy has enormous power in that these potential causes of sleep disturbances and sleep problems can be detected and objectified through combination with other instruments based on parental information (e.g., sleep diaries, sleep questionnaires, and symptom questionnaires). Sleep disorders can also be contributed to by other symptoms, such as pain [1]. In the clinical (but also the home) setting, actigraphy offers the potential to provide evidence of such influences by comparing the acquired data with symptom records from care providers, e.g., through the so-called 24-h sleep protocol. This may also increase the overall understanding and knowledge of how various symptoms interact with sleep in this complex sample and whether, in turn, there may be “typical” associations with, for instance, medication administration or dosing [57,58].

Furthermore, actigraphy allows an additional validation of external statements obtained, for example, by means of questionnaires. Parameters such as the number of nocturnal awakenings, for which a reliable statement is difficult to obtain by means of a proxy report, can thus be objectified [12,18,59].

Despite the overall satisfactory accuracy of actigraphy, the device might slightly underestimate and overestimate certain sleep parameters compared with polysomnography. This fact is less significant when practitioners are primarily concerned with a global assessment of specific sleep parameters (e.g., does the child wake up frequently during the night?). However, if very specific periods or times are involved (e.g., how many minutes was the child awake after falling asleep), one must be aware of this possible deviation and interpret and relativize the findings accordingly for the individual case.

Our findings, as outlined in other studies before, suggest that a reasonable configuration of the actigraph contributes significantly to the reliability of the generated sleep data [21,25,26]. Overall, a low or medium sensitivity level seems to be the best option for our population. It needs more data to enable meaningful conclusions to be drawn about which parameters are best captured by which sensitivity level. Nowadays, however, other devices from other manufacturers exist, and the applicability of conventional smartwatches is also increasingly being investigated [25,27,30,60]. Our results, especially with regard to the different sensitivity levels, could therefore be different if the corresponding measurements were taken with other devices. In the future, large comparative studies will be needed to test different devices in parallel against polysomnography in order to find out which specific device produces the most reliable data. 

### 4.1. Limitations

Despite the encouraging findings of our study, it has relevant limitations. First of all, the small sample size makes it difficult to transfer the results to the entire population of children with life-limiting neurological conditions. It may be that a repetition of the study with a larger sample would lead to different results than we were able to find. Because of the promising results, follow-up studies with a larger sample seem reasonable to also test the replicability of our results. Since the use of Bland-Altman plots is apparently well-suited for the comparison of PSG and actigraphy in such a complex sample as investigated in this study, this should be the method of choice for future sample design. Based on a study from 2016, the MedCalc program enables a sample size calculation considering the known expected mean of differences, expected standard deviation of differences and the maximum allowed difference between methods [61,62]. Thereby, it is necessary to take into account that different sample sizes may be required depending on the sleep parameters in focus. Referring to the results of this study, follow-up studies should, for example, aim for case numbers of N = 77–87 children if sleep onset and sleep offset of PSG and actigraphy are to be compared. Nonetheless, at this point, it is important to emphasize the difficile nature of the study design: For the group of severely ill children, which is small relative to other disabilities, polysomnography testing is, again, scheduled for only a very small proportion of patients. Although objectively, actigraphy hardly disturbs or affects the child, the application of such additional devices nevertheless increases the number of devices that the children carry with them anyway (e.g., ventilator). Against this background, the study had a rather high threshold for successful study participation, so N = 8 patients may well be deemed a success. Moreover, the authors do not claim that their findings are unrestrictedly generalizable. Rather, the results of this preliminary experimental study are promising and point to a field of research in pediatric palliative care that should be given more attention in the future in order to continuously improve diagnostics and therapy in this setting. In this regard, follow-up studies should also aim for a multicenter study design to increase the number of potential participants.

For the parameter sleep onset, the normal distribution assumption as a basis for the Bland-Altman plot could not be confirmed, so our results in this respect may theoretically only be interpreted cautiously. Nevertheless, we assume that this fact is due to the small sample and that a normal distribution could well be obtained with an increasing sample size. In view of the purpose of this study, which was to gather first insights into the applicability of actigraphy in a highly vulnerable population, this limitation is therefore justifiable and does not detract from the overall findings. 

The recording duration of our study was only one night. More meaningful data could be collected if actigraphy and polysomnography were compared over a longer period of time. However, in view of the complex study design described above and the high vulnerability of children with life-limiting diseases, a potential knowledge gain must always be carefully weighed against the resulting burden for the affected child. 

### 4.2. Conclusions

The basic applicability of actigraphy appears to be given even in a complex sample such as that of children with life-limiting neurological conditions who are cared for in pediatric palliative care. This has exciting implications for research and practice in that laborious and burdensome examinations such as polysomnography, while not being replaced, may potentially be usefully complemented in the future. The high burden on families could be reduced by making the examination of their child more pleasant and thus lead to an increased quality of life for those affected. Further, more comprehensive studies are necessary to evaluate the actual potential of actigraphy more precisely.

## Figures and Tables

**Figure 1 jcm-11-07107-f001:**
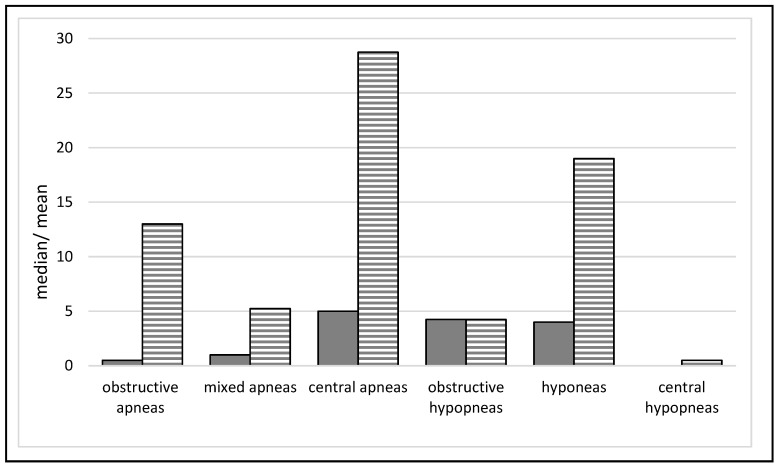
Median and mean (striped) frequency of respiratory events were assessed during the polysomnography (PSG) data collection night.

**Figure 2 jcm-11-07107-f002:**
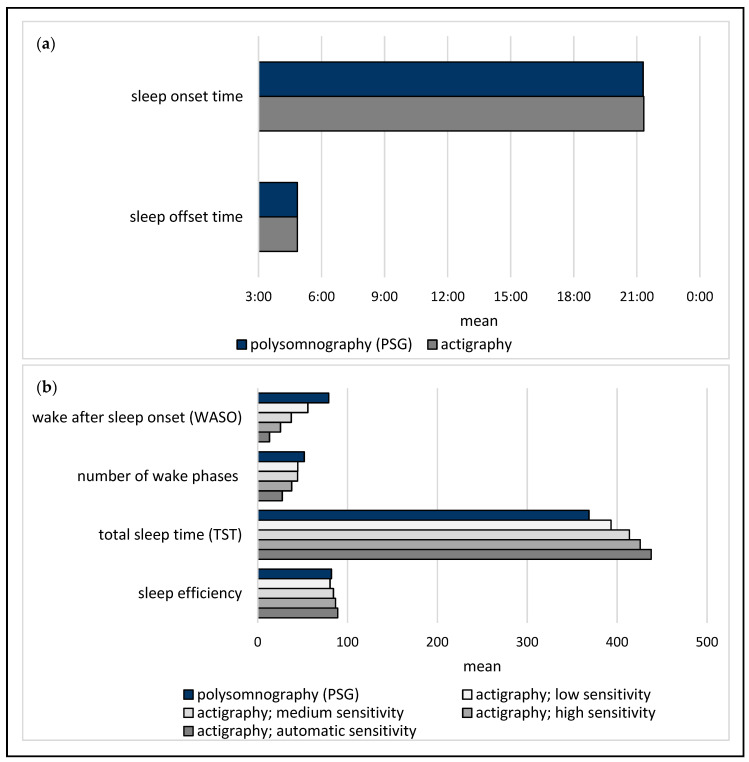
Mean polysomnography (PSG) and actigraphy values for the investigated parameters, for which no sensitivity levels (**a**) and those for which different sensitivity levels (**b**) were considered.

**Figure 3 jcm-11-07107-f003:**
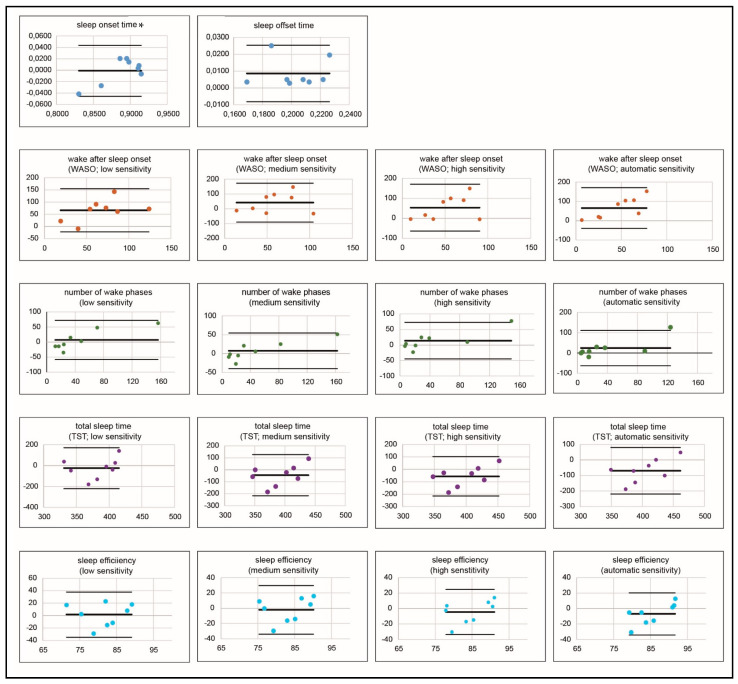
Bland-Altman plots for the investigated sleep parameters with the different sensitivity levels (the same color represents one sleep parameter each). * for this parameter, the normal distribution assumption could not be confirmed. The *x*-axis shows the mean value of the two measurement methods PSG and actigraphy, and the *y*-axis shows the difference between them. The differences between both measurement methods are entered as a horizontal bar in the diagram. The upper and lower limit are plotted as horizontal bars above and below the mean of the differences between the two focused measurement methods. If values are within the 95% confidence range, the second method can be said to have sufficient accuracy compared to the first [36].

**Table 1 jcm-11-07107-t001:** Patient characteristics.

Underlying Disease	Grouped (ICD-10 Code)	*n* (%)
Epileptic encephalopathy	disorder of the brain, unspecified (Q93.9)	1 (12.5)
Cerebral leukodystrophy	metabolic disorders (E70–E90)	1 (12.5)
Superior multisystem disease with foreground involvement of the central nervous system	congenital malformation of the brain, unspecified (Q04.9)	2 (25)
Joubert syndrome		
Arnold Chiari malformation type 2	other specified disorders of the brain (Q93.8)	1 (12.5)
	congenital malformation of the brain, unspecified (Q04.9)	1 (12.5)
Hypoxic-ischemic encephalopathy	other disturbances of the cerebral status of newborns (P91)	1 (12.5)
Trisomy 18	other specified congenital malformation syndromes affecting multiple systems (Q87)	1 (12.5)
Duchenne muscular dystrophy	primary disorders of muscles (G71)	1 (12.5)
care level ^1^		
1		0 (0)
2		1 (12.5)
3		2 (25)
4		1 (12.5)
5		4 (50)
born preterm		
Yes		3 (37.5)
No		5 (62.5)

**^1^** The care level (in Germany) expresses the degree to which a patient requires care ranging from 1 (slight impairment of independence) to 5 (most severe impairment of independence with special requirements for nursing care).

## Data Availability

Anonymized data are available from the corresponding author upon reasonable request.

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
