# Peer review of "Applicability of Actigraphy for Assessing Sleep Behaviour in Children with Palliative Care Needs Benchmarked against the Gold Standard Polysomnography"

_jcm, 2022, doi:10.3390/jcm11237107_

Round 1

Reviewer 1 Report

In the present study, a methodological comparison was performed regarding the gold standard polysomnography in relation to actigraphy in children receiving palliative care. For this group of patients it is of particular relevance whether the rather laborious and for the children stressful method of polysomnography for the investigation of sleep disorders can be replaced by actigraphy in the home environment.

The study provided promising results in this regard. However, due to the very small number of study participants (n=8), doubts arose about the statistical significance of the results. Although this aspect is mentioned in the limitations, no position is taken on the sample size planning for this or future studies. Accordingly, the following methodological questions arise for me:

 Over what time period were participants recruited? What was the target number of participants? Based on the available studies comparing actigraphy and polysomnography, were considerations made regarding the necessary sample size calculation? Is it possible to increase the number of cases?

Author Response

Applicability of actigraphy for assessing sleep behaviour in children with palliative care needs benchmarked against the gold standard polysomnography

Answers to Reviewer 1

In the present study, a methodological comparison was performed regarding the gold standard polysomnography in relation to actigraphy in children receiving palliative care. For this group of patients it is of particular relevance whether the rather laborious and for the children stressful method of polysomnography for the investigation of sleep disorders can be replaced by actigraphy in the home environment. The study provided promising results in this regard.

Thank you for this recognition and the opportunity to revise our paper.

However, due to the very small number of study participants (n=8), doubts arose about the statistical significance of the results. Although this aspect is mentioned in the limitations, no position is taken on the sample size planning for this or future studies. Accordingly, the following methodological questions arise for me:

 Over what time period were participants recruited? What was the target number of participants? Based on the available studies comparing actigraphy and polysomnography, were considerations made regarding the necessary sample size calculation? Is it possible to increase the number of cases?

Thank you for these helpful comments. We have added the relevant information in the methods section and the discussion:

  • 3, l. 143: “Data collection took place from January 2021 to March 2022.”
  • 3, ll. 101-105: “A small number of cases was considered appropriate for sampling considering the intricate conditions of study inclusion respectively the small achievable population. For its definition, studies with a similar design and children with complex underlying dis-eases were considered, ranging from N=9-11 children [29,32,33] , so that the target number of children for this study was set at approximately N=10.”
  • 10-11, ll. 432-443: “Because of the promising results, follow-up studies with a larger sample seem reasonable to also test the replicability of our results. Since the use of Bland-Altman plots is apparently well-suited for the comparison of PSG and actigraphy in such a complex sample as investigated in this study, this should be the method of choice for future sample design. Based on a study from 2016, the MedCalc program enables a sample size calculation considering known expected mean of differences, expected standard deviation of differences and the maximum allowed difference between methods [62,63]. Thereby, it is necessary to take into account that different sample sizes may be required depending on the sleep parameters in focus. Referring to the results of this study, follow-up studies should, for ex-ample, aim for case numbers of N=77-87 children if sleep onset and sleep offset of PSG and actigraphy are to be compared.”

p. 10, ll. 454-455: “In this regard, follow-up studies should also aim for a multicenter study design to in-crease the number of potential participants

Reviewer 2 Report

Thank you for the opportunity to review this interesting study and valuable study.

I acknowledge that the significant actigraphy aportations are reported and the discussion does address the few literatures available. Still, readers need to know the relevance of the results could help to expand the possibilities of objective sleep diagnostics, what would impact in quality of life, activities of daily living, caregivers, and care in children with palliative care needs.

Although it is established that actigraphy can provide additional information on sleep behavior in children with respect to the gold standard, it is necessary to further argue the discussion. It is suggested to discuss the contributions of actigraphy with respect to PSG.

Many potential causes that need to be considered including physical health conditions such as upper airway obstruction; pain leading to poor sleep, and environmental factors.  You need to reflect on the implications of these results, make suggestions about how

that its results could be used to improve practice and maybe make recommendations

aimed at increasing knowledge in that field. It is recommended to review the results discussed, emphasizing these:

- line 335: In detail, our patients spent more time in sleep stages than those with mental retardation, but less in the REM sleep stage…

- Line 341: However, the sleep parameters in all these studies and in our study appear to differ from those of healthy children..

- Line 348. Apneas and hypopneas were the most common respiratory abnormalities in our sample. This finding fits with existing literature, which also showed a predominance of these abnormalities over other respiratory problems…

Author Response

Applicability of actigraphy for assessing sleep behaviour in children with palliative care needs benchmarked against the gold standard polysomnography

Answers to Reviewer 2

Thank you for the opportunity to review this interesting study and valuable study.

I acknowledge that the significant actigraphy aportations are reported and the discussion does address the few literatures available. Still, readers need to know the relevance of the results could help to expand the possibilities of objective sleep diagnostics, what would impact in quality of life, activities of daily living, caregivers, and care in children with palliative care needs.

Although it is established that actigraphy can provide additional information on sleep behavior in children with respect to the gold standard, it is necessary to further argue the discussion. It is suggested to discuss the contributions of actigraphy with respect to PSG.

 Thank you for this recognition and the opportunity to revise our paper.

Many potential causes that need to be considered including physical health conditions such as upper airway obstruction; pain leading to poor sleep, and environmental factors.  You need to reflect on the implications of these results, make suggestions about how that its results could be used to improve practice and maybe make recommendations aimed at increasing knowledge in that field. It is recommended to review the results discussed, emphasizing these:

- line 335: In detail, our patients spent more time in sleep stages than those with mental retardation, but less in the REM sleep stage…

- Line 341: However, the sleep parameters in all these studies and in our study appear to differ from those of healthy children..

- Line 348. Apneas and hypopneas were the most common respiratory abnormalities in our sample. This finding fits with existing literature, which also showed a predominance of these abnormalities over other respiratory problems…

Thank you very much. According to your helpful comments, we have further elaborated the discussion:

  • 9, ll. 354-365: “More specifically, according to the International Classification of Sleep Disorders, various sleep disorders are associated with a change in sleep architecture. For example, sleep related hypoventilation may be associated with decreased sleep stages 3 and REM, disorders of arousal with altered microstructure associated with increased slow wave activity during NREM sleep cycles [43]. However, accurate diagnosis and interpretation of these sleep disorders requires knowledge of which "baseline" changes in sleep architecture may be associated with various underlying diseases and thus can be distinguished from changes that are "purely" associated with corresponding sleep disorders. Such knowledge allows not only optimization of diagnosis, but also of therapy for children with life-limiting neurological conditions, since a more targeted decision can be made as to which sleep disorders can potentially be cured and which can merely be regulated to an acceptable level.”
  • 9, ll. 368-374: “Breathing problems are a common symptom, especially at the end of life in critically ill children. Diagnosis as early and conclusively as possible can help alleviate corresponding suffering in patients and families [47]. Knowledge of which respiratory problems occur how frequently in this population also enables the anticipatory development of different therapy strategies and may thus substantially improve the quality of life of children with life-limiting neurological conditions.”
  • 10, ll. 391-405: “Although it seems reasonable that sleep disorders in children with life-limiting neurological conditions are mainly caused by primary disruptions of sleep-regulating processes in the brain associated with the underlying disease, it should not be neglected that they may (also) be promoted by other factors such as environmental influences like noise, or inadequate sleep hygiene [12,54-56]. Especially in the home setting, actigraphy has enormous power in that these potential causes of sleep disturbances and sleep problems can be detected and objectified through combination with other instruments based on pa-rental information (e.g. sleep diaries, sleep questionnaires, symptom questionnaires). Sleep disorders can also be contributed to by other symptoms such as pain [57]. In the clinical (but also the home) setting, actigraphy offers the potential to provide evidence of such influences by comparing the acquired data with symptom records from care providers, e.g., through the so-called 24-hours sleep protocol. This may also increase the overall understanding and knowledge of how various symptoms interact with sleep in this complex sample and whether, in turn, there may be “typical” associations with, for instance, medication administration or dosing [58,59].

Round 2

Reviewer 1 Report

Thank you for your response and valuable changes.

Reviewer 2 Report

The authors did a good job responding to comments.